# Does frequency of supportive supervisory visits influence health service delivery?—Dose and response study

Binyam Fekadu Desta[1]*, Ismael Ali Beshir[1], Bekele Belayhun Tefera[2], Mesele Damte Argaw[1], Habtamu Zerihun Demeke[2], Mengistu Asnake Kibret[2]

1 USAID Transform: Primary Health Care project, JSI Training & Research Institute, Inc., Addis Ababa, Ethiopia, 2 USAID Transform: Primary Health Care project, Pathfinder International Ethiopia, Addis Ababa, Ethiopia

* binyamfe@gmail.com, binyam_desta@jsi.com

**Data Availability Statement:** All relevant data are within the manuscript and its Supporting Information files.

## Abstract

High quality care—at a minimum—is a combination of the availability of tangible resources as well as a capable and motivated health workforce. Researchers have suggested that supportive supervision can increase both the performance and motivation of health workers and the quality of care. This study is aimed at assessing the required number of visits and time between visits to bring about improvements in health service delivery. The study employed a primary health care performance improvement conceptual framework which depicts building blocks for improved health service delivery using longitudinal program outcome monitoring data collected from July 2017 to December 2019. The analysis presented in this study is based on 3,080 visits made to 1,479 health centers in the USAID Transform: Primary Health Care project's intervention districts. To assess the effects of the visits on the repeated measure of the outcome variable (Service-Delivery), multilevel linear mixed model (LMM) with maximum likelihood (ML) estimation was employed. The results showed that there was a significant dose-response relationship that consistent and significant improvement on Service-Delivery indicator was observed from first ($\beta$ = -26.07, t = -7.43, p < 0.001) to second ($\beta$ = -21.17, t = -6.00, p < 0.01), third ($\beta$ = -15.20, t = -4.49, p < 0.02), fourth ($\beta$ = -12.35, t = -3.58, p < 0.04) and fifth ($\beta$ = -11.18, t = -2.86, p < 0.03) visits. The incremental effect of the visits was not significant from fifth visit to the sixth suggesting five visits are the optimal number of visits to improve service delivery at the health center level. The time interval between visits also suggested visits made between 6 to 9 months ($\beta$ = -2.86, t = -2.56, p < 0.01) showed more significant contributions. Therefore, we can conclude that five visits each separated by 6 to 9 months elicits a significant service delivery improvement at health centers.

## Introduction

Since its adoption, primary health care has valued the role of health providers and quality of care. In its renewal for commitment, the Astana declaration clearly states the need for

**Funding:** It is funded by United States Agency for International Development (USAID) under cooperative agreement number of AID-663-A-17-00002. The authors' views expressed in this study report do not necessarily reflect the views of USAID or the United States Government. The funder provided support in the form of salaries to authors [BFD, IAB, BBT, MDA, HZD, MAK], but did not have any additional role in the study design, data collection and analysis, decision to publish, or preparation of the manuscript. The specific roles of these authors are articulated in the 'author contributions' section.

**Competing interests:** The authors have declared that no competing interests exist. The authors' affiliation does not alter their adherence to PLOS ONE policies on sharing data and materials.

competent health providers in high quality health care. High quality care—at a minimum—is a combination of the availability of tangible resources as well as a capable and motivated health workforce [1]. Researchers have suggested that supervision can increase both the performance and motivation of health workers and the quality of care [2, 3, 4]. This is reinforced by the introduction of supportive supervision as part of service improvement initiatives in six countries—Bangladesh, Brazil, Honduras, Kenya, Nepal, and Tanzania—who have yielded promising results in both service quality and providers' performance [3].

Morrison defines supervision as, "...a process by which one worker is given responsibility by the organization to work with another worker(s) in order to meet certain organizational, professional and personal objectives" [5]. Supervision is believed to be a collaborative platform where the supervisee offers an honest and open account of their work, and the supervisor offers feedback and guidance to improve performance and quality of care [6]. When the supervision is supportive, it intends to observe the health care actions of the provider, provide feedback from the supervisor to the provider on performance, and establish collaborative problem solving to improve performance [1]. Usually tools such as checklists, job aids, guidelines and, to some extent, mobile technology or e-Health devices are used to facilitate data collection, identification of problems and record-keeping [7]. However, the use of guidelines and checklists for the supportive supervision process, may not be enough to effect changes in performance [3]. Any supportive supervision hence requires, a) good knowledge of the local situation; b) opportunity for the supervisor and supervisee to work together on the issue; c) frequent constructive feedback; and d) structured or scheduled supervision with agreed content and learning [7].

Although there is considerable literature on supervision, there is limited literature on the outcomes; such as providers' competence, improvements in quality of care and service utilization, associated with supervision [6]. The available literature also fails to identify the optimal amount and timing of supervisions [8]. USAID Transform: Primary Health Care, a USAID funded project supporting the government of Ethiopia in health Sector Transformation Plan and preventing child and maternal deaths, implements supportive supervision to bring about changes in the health system's performance as well as quality of care. This study is thus aimed at assessing the required number of visits and the ideal interval between visits to bring about changes in health service delivery as well as identify project related factors contributing to the effectiveness of supervisions.

## Materials and method

### Study settings

USAID Transform: Primary Health Care covers a total of 396 districts in the four largest regions of Ethiopia (Amhara, Oromia, SNNP, and Tigray) where a total of 1,880 health centers provide health care to 53 million people. A health center is a health facility at the primary level of the health care system which provides promotive, preventive, curative and rehabilitative outpatient care including basic laboratory and pharmacy services with a capacity for 10 beds for emergency and delivery services. It is staffed with medical doctors, BSc as well as diploma level health science graduates including clinical officers, nurses, midwives, and lab technicians. On average a health center can have 35 direct service providers, and support staff [9]. On average, a health center is designed to provide health care services to 25,000 people residing in its catchment area.

### Intervention

A supportive supervision checklist is a set of questions related to reproductive, maternal and child health and health system interventions which was developed by the USAID Transform:

Primary Health Care project to guide field level support. The checklist is organized to frame a two-way discussion between the supervisor and the health worker at each institution. Each question has a definition, decision point and a response documentation section for improvement plan. The supervisors responsible for conducting the supervisions and providing technical guidance are—at a minimum- a first degree graduates in health studies, have experience of working at the primary level of care, and have attended a supervision technique training. During each visit, a supervisor is expected to spend at least half a day in the facility. When a supervisor goes to the institutions, s/he is expected to follow the checklist and record the findings and work with the staff and management of the health facility to bring about improvements on the identified problems.

## Data collection

During facility support, data collection and entry is conducted onsite using an online electronic system and tablets. The system allows the questionnaires to be programmed and follow skip patterns based on previous responses. On a few occasions, the visit may be carried out by other experts who will use a paper format and then transfer the data to the online system.

## Study design and instruments

The study employed a retrospective cohort study. For assessment purposes, a primary health care performance improvement conceptual framework for primary health care (Fig 1) was used to categorize the questions into the major domain. The framework considers the role of service organization and quality of care as important drivers for primary health care performance [10]. As the supportive supervision is targeted to improve service delivery and its management, the Service-Delivery component of the framework was considered. A total of 30 questions were categorized into the five Service-Delivery components—access, availability of effective PHCs, high quality primary health care, population health management, and facility organization and management.

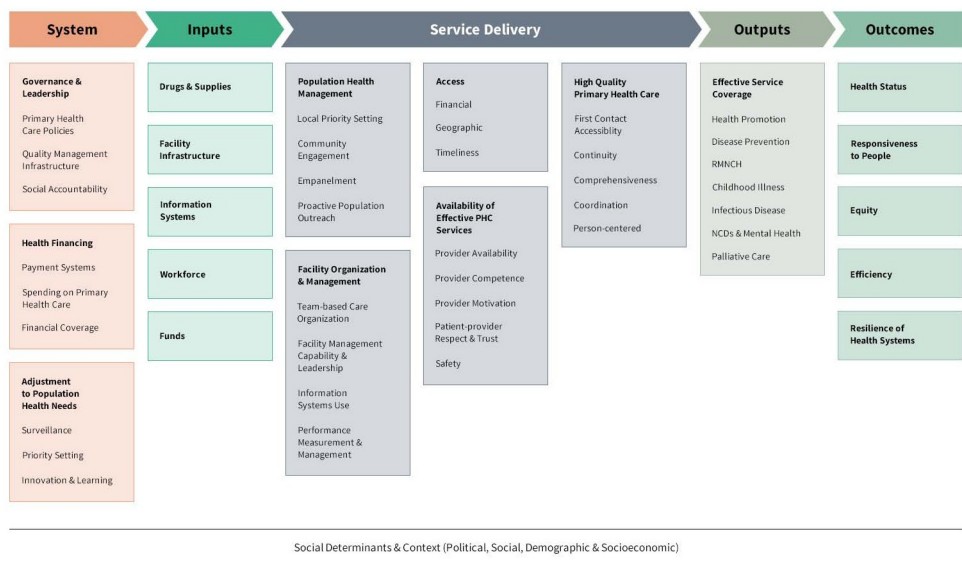

**Fig 1. Primary Health Care Performance Improvement (PHCPI) conceptual framework for primary health care.**

## Data source

The study uses a longitudinal program outcome monitoring data collected from July 2017 to December 2019. The USAID Transform: Primary Health Care project monitoring data is collected from the project intervention woreda health offices, primary hospitals, health centers, health posts, and households during routine and random supportive supervision visits with the objective of providing onsite technical support and producing unbiased data for decision making. During this period, a total of 1,322, 499, 3,080, 4,741, and 23,151 visits were made to woreda health offices, primary hospitals, health centers, health posts, and households respectively. The analysis presented in this study is based on the 3,080 supportive supervision visits made to 1,479 health centers in the project's intervention districts.

## Types of variables

**Outcome variables.** The composite measure of the service delivery of primary health care which was the average of the five Service-Delivery components of the PHCPI framework—access, service availability, patient centered care, population health management, and service organization and management—was considered as the outcome variable (Service-Delivery). A high score of this variable suggested the availability of better facility services.

**Exposure variables.** The number of visits to health facilities and the interval between consecutive visits were accounted as exposure variable for this study.

**Control variables.** The study had two levels of control variables. The first group includes the organization of the project support structure—facility distance from cluster office (CLO), average number of woreda per cluster staff, number of low performing woreda in the CLO and region, and the second level was related to health facility factors—number of technical staff, facility infrastructure (water and electricity), catchment population size, facility distance from woreda capital, and facility head's experience in years.

**Data analysis.** Data were managed using a web-based system, DHIS2 [11], and exported to SPSS version 25 for statistical analysis. Both descriptive and inferential statistics were applied. Descriptive statistics were used to analyze the five service-delivery components. To assess the effect of the control variables on the repeated measure of the outcome variable (service-delivery), multilevel linear mixed model (LMM) with maximum likelihood (ML) estimation was employed. In addition, the effects of access to roads on frequency of visits was also tested using multinomial logistics regression.

Since the data had unequal sample sizes, inconsistent time interval, and missing data, applying univariate and multivariate tests of statistics was not recommended [12]. LMM is an appropriate approach when studying individual change as it creates a two-level hierarchical model that nests time within individual [13]. In addition, the study's interest was on the subject-specific (facilities) interpretation of effects and identifying group variance sources, therefore, LMM was preferred over a generalized estimation equation to fit the data. The overall effect of each control variable on the Service-Delivery was tested through an F-test, while the effect of each category of each factor was tested through t-test with the respective degrees of freedom.

To determine the best fit model, first, an unconditional mean model was used. In this model, no predictor was included. This model served as a baseline model to examine individual variation in the outcome variable without regard to time [14]. The model assesses the differences between the observed mean value of each facility and the true mean from the population. If the variation is high, it suggests that certain amount of outcome variation could be explained by the predictors at that level. Then a model containing time (number of visits) as a fixed and random effect was applied. This model tests if time (number of visits) is significant

by examining the presence of interindividual difference in trajectory change over time. Finally, a model containing the fixed effects of variables of interest, the random intercept, and the random slopes were fitted.

To select the best model, -2 log likelihood ratio test and Bayesian Information Criterion (BIC) were used. Generally, the smaller the statistical value, the better the model fit into the data. In all the statistical tests, significance was refereed at $p < 0.05$.

**Ethical considerations.** The study considered aggregate secondary program data. The JSI Institutional Review Board (IRB) has determined that the study does not constitute "human subjects research" under US HHS regulation 45 CFR 46.102(f).

## Results

The study results are presented in three sections: 1) characteristics of the study facilities, 2) description of the Service-Delivery, the outcome variable, and 3) the multilevel linear mixed model (LMM) analysis.

### Characteristics of the study facilities

Overall, 1,479 heath centers were included in the study (Table 1). All the facilities had received at least one visit (100%), during the study period. Of these facilities, 889 (28.9%) received two visits, 438 (14.2%) received three visits, 165 (5.4%) received four visits, and 105 (3.4%) received

**Table 1. Characteristics of the study facilities and description of visits.**

| Characteristics | Number (percent) |
|---|---|
| **Facility distribution by region (n = 1479)** | |
| Amhara | 417 (28.2) |
| Oromia | 614 (41.5) |
| SNNP | 318 (21.5) |
| Tigray | 130 (8.8) |
| **Facility infrastructure availability (n = 1479)** | |
| Water | 876 (59.2) |
| Electricity | 1063 (71.9) |
| **Facilities with access to roads (n = 1479)** | 1378 (93.2) |
| **Facility head years of experience (n = 1466)** | |
| <= 1 Year | 641 (43.7) |
| 1–3 Years | 507 (34.6) |
| 3–5 Years | 172 (11.7) |
| > 5 Years | 146 (10.0) |
| **Number of visits (n = 3076)** | |
| 1st visit | 1479 (48.1) |
| 2nd visit | 889 (28.9) |
| 3rd visit | 438 (14.2) |
| 4th visit | 165 (5.4) |
| 5+ visits | 105 (3.4) |
| **Duration between consecutive visits (n = 3076)** | |
| Visited between 3 months | 1797 (58.4) |
| Visited between 3–6 months | 414 (13.5) |
| Visited between 6–9 months | 300 (9.8) |
| Visited between 9–12 months | 264 (8.6) |
| Visited after 12 months | 301 (9.8) |

five or more visits. The maximum number of visits to a facility during this period was seven visits. From the total facilities visited, 417 (28.2%), 614 (41.5%), 318 (21.5%), and 130 (8.8%) facilities were in Amhara, Oromia, SNNP, and Tigray regions respectively. The average number of days between visits was 119.3±161.0 standard deviation (SD) days, approximately four months. The average distance from a facility to the woreda's capital was 107.3±40.7 SD kilometers. Almost all 1,381 (93.4%) facility woredas were located within 50 kilometers distance from the project's cluster offices. The study also tested the influence of access to roads on the number of supervisory visits and found no significant relationship.

## Description of the service-delivery

Table 2 shows a consistent dose-response relationship between the number of visits and the 30 questions that are categorized to form the five Service-Delivery components. During the first round of visits, the facilities' performance coverage was 62, 49, 54, 51, and 39 and improved to 78, 80, 75, 68, and 52 in the fifth and above visits for access, patient centered care, service organization and management, service availability, and population health management respectively. Similarly, the average Service-Delivery performance increased from 49.9 (at the first visit) to 69.0 (at 5+ visits) (Fig 2).

Relatively as low as 3.9, 4.1, and 3.1 average percentage change between the visits were observed in access, service availability, and population health management compared to patient centered care and service organization and management, which were 7.6 and 5.4 respectively. A positive effect was observed regarding visit frequency between first, second and third visits for all the Service-Delivery components. However, the effect of visit frequency between fourth and fifth visits is not positive for all the components as there was a slight decrease of 1.1 and 0.2 for service availability and population health management performances respectively.

## Results from multilevel linear mixed model (LMM)

The Intra-class Correlation Coefficient (ICC) was 85.68/ (85.68 + 229.04) = 0.27, indicating that about 27.2 percent of the total variation was due to interindividual differences. The value was greater than the minimum recommended value of 25 percent and suggested using a mixed model for the data [15]. The estimates of covariance parameters, SPSS output is shown in Table 3.

After the null model test, the next model fitted was the unconditional linear growth curve model containing time (number of visits) as a fixed and random effect. Accordingly, the resulting output showed a significant linear increase in the Service-Delivery ($\beta = 5.00$, SE = 0.28, $p < 0.001$). The mean estimated initial status was 45.32 and the linear growth rate was 5.00 (Table 4). This suggested that the mean Service-Delivery indicator was 45.32 and increased with time. The random error terms associated with the intercept and linear effect were also significant ($p < 0.001$).

A comparison of models 1 and 2 showed a decline of 33.39 (229.042 to 195.651) in the residual variance. This indicated that about 33.4 percent of the linear rate of change in the Service-Delivery indicator was associated with number of visits.

Finally, a model containing both levels, project support structure and health facility factors, of the control variables as fixed effects, number of visits (time) as a repeated effect, and duration between consecutive visits as a random effect was fitted to explore group differences in change over time.

Accordingly, the fixed intercept, duration between consecutive visits, infrastructure (electricity and water), region, facility's distance from the woreda capital, woreda's distance from

**Table 2. Service-delivery components trend.**

| Proportion of facilities | % | | | | |
|---|---|---|---|---|---|
| | 1st Visit | 2nd Visit | 3rd Visit | 4th Visit | 5+ Visits |
| **Provide all exempted health services free of charge** | 90.8 | 93.0 | 94.4 | 96.0 | 97.1 |
| **Provide health care services to CBHI beneficiaries** | 72.5 | 78.4 | 77.7 | 79.5 | 90.6 |
| **Access to roads** | 94.2 | 95.1 | 95.4 | 96.9 | 98.1 |
| **Has at least one ambulance** | 24.0 | 26.2 | 29.7 | 32.1 | 36.2 |
| **Access** | 62.4 | 66.0 | 68.6 | 69.1 | 78.1 |
| **Trained staff use chart booklets while providing services** | 74.4 | 78.2 | 81.8 | 90.2 | 84.2 |
| **Delivery partograph is used correctly** | 64.4 | 73.2 | 76.2 | 79.3 | 83.0 |
| **Under-five children classified correctly** | 65.9 | 66.8 | 71.4 | 77.8 | 82.4 |
| **Under-five children treated correctly** | 59.3 | 61.3 | 65.5 | 72.5 | 79.4 |
| **Patient centered care** | 49.4 | 63.8 | 71.3 | 77.4 | 79.8 |
| **Reviewed and reported EHCRIG chapters in the most recent quarter** | 65.4 | 81.0 | 91.7 | 89.8 | 97.1 |
| **Followed IPLS standards to ensure uninterrupted supply chain** | 68.5 | 75.8 | 81.3 | 89.0 | 92.4 |
| **Used HMIS data for planning and decision making** | 70.3 | 74.0 | 82.5 | 86.5 | 88.6 |
| **Used LQAS for data accuracy check** | 69.1 | 71.8 | 77.7 | 85.1 | 86.4 |
| **HC Director trained on Leadership, Management and Governance (LMG)** | 19.0 | 22.0 | 26.2 | 34.0 | 42.3 |
| **Established case review/audit system for maternal and newborn death** | 41.4 | 47.9 | 51.9 | 55.9 | 70.3 |
| **Have an EPI defaulter tracing mechanism** | 65.8 | 73.2 | 76.0 | 82.9 | 82.4 |
| **Established a QI team and assigned a focal person for QA/QI** | 46.0 | 49.3 | 56.4 | 62.0 | 54.3 |
| **Service organization and management** | 53.6 | 58.2 | 65.1 | 68.9 | 75.0 |
| **All expected FP methods are available in all days in the past one month** | 60.7 | 63.9 | 68.1 | 75.5 | 71.8 |
| **PPFP service is available in delivery room** | 34.3 | 45.8 | 49.8 | 58.7 | 65.4 |
| **Provided all BEmONC signal functions** | 57.8 | 68.9 | 74.9 | 82.4 | 86.1 |
| **Provided women friendly delivery services** | 78.2 | 85.8 | 90.0 | 92.8 | 94.0 |
| **Provided ferrous sulfate for pregnant women during ANC** | 87.0 | 92.0 | 92.0 | 95.3 | 92.6 |
| **Functional maternity waiting room/home** | 73.0 | 71.7 | 73.6 | 81.2 | 73.1 |
| **ANC clients tested for syphilis** | 53.1 | 61.8 | 69.5 | 68.2 | 67.7 |
| **Mothers received Uterotonics in the third stage of labor or immediately after birth** | 73.8 | 82.4 | 89.5 | 92.2 | 88.8 |
| **Newborns received newborn care** | 67.5 | 68.9 | 75.0 | 84.2 | 84.3 |
| **Newborns with neonatal sepsis received treatment** | 70.5 | 76.8 | 73.8 | 86.7 | 75.0 |
| **Asphyxiated newborns resuscitated** | 89.7 | 94.9 | 97.0 | 96.3 | 97.5 |
| **Service availability** | 51.4 | 59.5 | 63.9 | 69.0 | 67.9 |
| **Exercise community feedback collecting mechanisms/town hall meetings** | 31.6 | 33.7 | 38.8 | 40.1 | 47.1 |
| **Have a social behavior change communication plan** | 35.8 | 35.1 | 42.8 | 48.0 | 44.1 |
| **Work together with kebele administration** | 73.1 | 70.8 | 72.7 | 80.5 | 79.6 |
| **Population health management** | 39.4 | 41.8 | 47.0 | 52.0 | 51.7 |
| **Service-Delivery** | 49.9 | 55.7 | 61.3 | 64.4 | 69.0 |

the cluster office, average number of woreda per cluster staff, and number of visits (time) were statistically significant (p value < 0.05). However, facility head's experience in years, number of technical staff in the facility, catchment population size and number of low performing woredas in the CLO were not found to be significant or independent predictors of the Service-Delivery outcome variable. Table 5 shows the respective F-test values and exact p values.

The estimates of fixed effects table, Table 6, gives the same p values including estimates of the group effect sizes and the 95 percent confidence intervals for the estimates. Only number of visits (time) and duration between consecutive visits is shown. The comparison of the categories revealed that there was a significant dose-response relationship between the number of

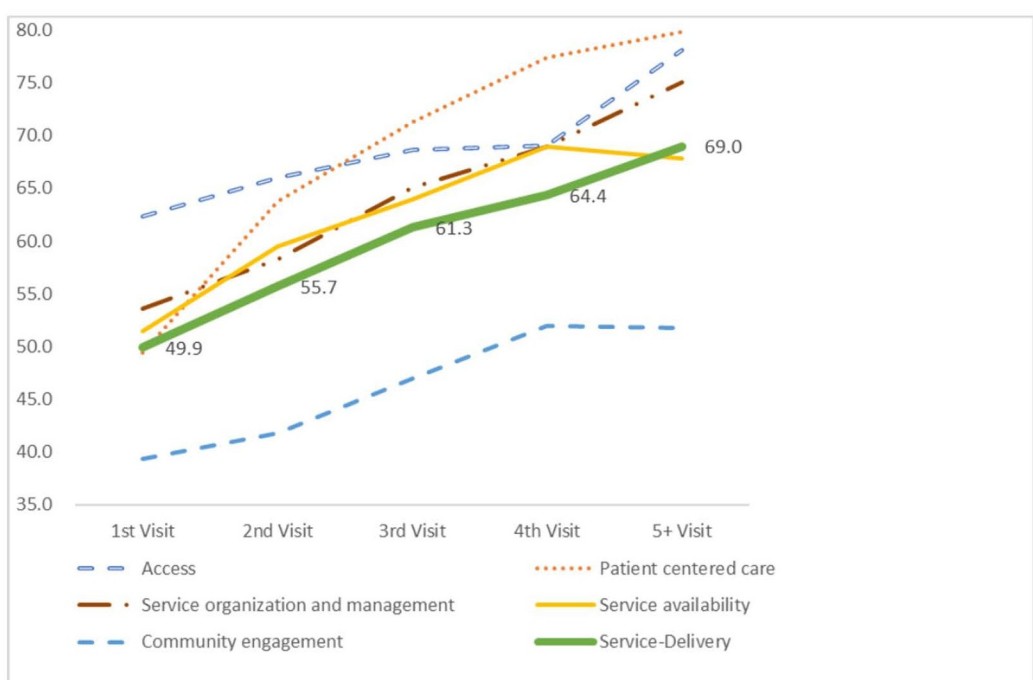

**Fig 2. Trend of mean service-delivery and service-delivery components.**

**Table 3. Variability of intercepts of the null model, unconditional mean model.**

| Parameter | Estimate | 95% CI | | P value |
| --- | --- | --- | --- | --- |
| | | Lower | Upper | |
| Residual | 229.042 | 214.317 | 244.778 | <0.001 |
| Variance for intercept [subject = Code] | 85.679 | 71.079 | 103.279 | <0.001 |

visits and the Service-Delivery indicator. A consistent and significant improvement on Service-Delivery indicator was observed from first ($\beta$ = -26.07, t = -7.43, p < 0.001) to second ($\beta$ = -21.17, t = -6.00, p < 0.01), third ($\beta$ = -15.20, t = -4.49, p < 0.02), fourth ($\beta$ = -12.35, t = -3.58, p < 0.04) and fifth ($\beta$ = -11.18, t = -2.86, p < 0.03) visits. The incremental effect of the visits was not significant going from fifth visit to sixth, suggesting five visits are the optimal number of visits to improve the Service-Delivery components of a health center. Similarly, the duration between consecutive visits showed a significant improvement on facilities visited between 3 months ($\beta$ = -4.20, t = -3.82, p < 0.001), 3 to 6 months ($\beta$ = -4.01, t = -3.78, p < 0.001), and 6 to 9 months ($\beta$ = -2.86, t = —2.56, p < 0.01). The timing of visits also suggested visits made between 3 to 6 months produced smaller changes compared to visits made

**Table 4. Estimates of fixed effects.**

| Parameter | Estimate | SE | 95% CI | | p value |
| --- | --- | --- | --- | --- | --- |
| | | | Lower | Upper | |
| Intercept | 45.320 | .584 | 44.173 | 46.466 | <0.001 |
| Number of visits (Time) | 4.995 | .283 | 4.439 | 5.551 | <0.001 |

**Table 5. Tests of fixed effects on service-delivery.**

| Source | F | P value |
|---|---|---|
| Intercept | 1190.106 | <0.001 |
| Number of visits | 38.017 | <0.001 |
| Duration between consecutive visits | 4.734 | .001 |
| Facility head's experience in years | 1.673 | .171 |
| Electricity | 25.100 | <0.001 |
| Water | 13.942 | <0.001 |
| Region | 41.795 | <0.001 |
| Facility number of technical staff | 2.990 | .084 |
| Facility catchment population size | .536 | .464 |
| Facility distance from woreda capital | 36.297 | <0.001 |
| Facility distance from cluster office | 6.078 | .014 |
| Avg. number of woreda per cluster staff | 12.052 | .001 |
| # of low performing woredas in the CLO | .604 | .437 |

**Table 6. Estimates of fixed effects on service-delivery.**

| Parameter | Coefficient | df | t | 95% CI | | p value |
|---|---|---|---|---|---|---|
| | | | | Lower | Upper | |
| Intercept | 88.52 | 6.58 | 21.59 | 78.70 | 98.35 | <0.001 |
| Number of visits | | | | | | |
| 1st visit | -26.07 | 3.66 | -7.43 | -36.18 | -15.97 | <0.001 |
| 2nd visit | -21.17 | 3.58 | -6.00 | -31.45 | -10.90 | 0.01 |
| 3rd visit | -15.20 | 3.21 | -4.36 | -25.90 | -4.49 | 0.02 |
| 4th visit | -12.35 | 2.85 | -3.58 | -23.68 | -1.02 | 0.04 |
| 5th visit | -11.18 | 6.19 | -2.86 | -20.69 | -1.68 | 0.03 |
| 6th visit | -10.78 | 1.26 | -2.89 | -40.45 | 18.90 | 0.17 |
| 7th visit | 0 | | | | | |
| Duration between consecutive visits | | | | | | |
| Visited between 0–3 months | -4.20 | 553.88 | -3.82 | -6.37 | -2.04 | <0.001 |
| Visited between 3–6 months | -4.01 | 963.83 | -3.78 | -6.09 | -1.93 | <0.001 |
| Visited between 6–9 months | -2.86 | 1135.72 | -2.56 | -5.05 | -0.67 | 0.01 |
| Visited between 9–12 months | -2.22 | 1185.02 | -1.92 | -4.49 | 0.05 | 0.06 |
| Visited after 12 months | 0 | | | | | |

between 6 to 9 months. The beta values of the estimated fixed effects reported are negative as the last visit was used as a reference.

## Discussion

The supportive supervision provided at primary health care is an effective tool and best utilized through the guidance of job-aids or checklists, a process of joint problem solving and further follow-up on agreed points [3]. The use of primary health care evaluation model helped the researchers to organize questions and findings and measure the contribution of supervisions to improvements in the health service delivery. The model's comprehensiveness is thus helpful for the development of supervision tools and to conduct similar studies that measure the contribution of investments in improving primary health care.

The success of implementation of supportive supervision depends on regularity of supervisory visits to health facilities to build relationships, monitor performance, and develop skills of problem solving among the team involved in the supervision [16]. Studies conducted in various settings showed improvements in various dimension of health service delivery [16]. For example, a study conducted in Tanzania using an electronic checklist demonstrated a statistically significant increase of 3–7 percent in mean score of performances within primary health care settings [17]. This study also found that the average score of Service-Delivery, based on the five components of the model, showed significant improvements as the number of visits increases from 49.9 (at first visit) to 69.0 (at 5+ visits). However, the observed changes at each subcomponent of the evaluation model is different.

The major changes in the Service-Delivery components were observed on patient centered care (from 49 to 80%—31 points) and service organization and management (54 to 75%—21 points). Like this finding, various studies underlined the important role of supervision to improving service quality. The studies reported that supervision enhances compliance with processes, and adherence to standards and guidelines that are associated with enhanced patient health outcomes in South Africa, India, and Bangladesh [4, 18]. In other studies, the activity contributes to improvements in medicine management and treatment of common childhood illnesses [7]. This can be explained by the fact that indicators included in these two components can be improved by providing major mentorship support and availing the required management related resources at the health centers. The investments and quality checks made regularly in the supervision processes also contributed to developing the skills of supervisors to improve their communication skills and understand the context and technical skills on the contents of the supervision checklists which as mentioned by various literatures, were good attributes for observed changes in these dimensions.

In contrast, population health management contributed the lowest to the overall changes. This can be explained by the fact that a significant proportion of community engagement activities are driven by health extension workers placed at the village level. Therefore, the health centers may not have the necessary documentation to show progresses on the indicators included in this category. The other two categories—access and service availability—bring with them a fair amount of contributions to the overall changes. Similar to these findings, studies conducted in Tanzania showed that improvements in clinical practice and facility administration and management were slightly less marked [17] and there was no effect on availability of basic equipment among the health facilities across the six integrated supportive supervision visits [16]. This is because majority of the indicators require huge investment and resources and require changes in the policy environment. In addition, some of the indicators are far from the power and circle of influence of the supervisors going to the facilities. These findings underscored the importance of various levels of engagement and different interventions which include the health workers and decision makers at various levels of the health administration.

Frequency of visits and duration between visits are very important factors for the observed changes. A supervision visit may take two forms: comprehensive or issue specific. The study highlights the optimum number of visits to influence all components of health service delivery. All components of the Service-Delivery indicators increased with visit frequencies up to the fourth visit. However, some components decreased after the fourth visit. Any supervisory visit after the fourth visit should thus consider issue specific support. In addition, visits which took place before nine months did demonstrate changes in performances. However, further changes in performance were observed when the duration between the visits was 6 to 9 months. Studies conducted in various countries also demonstrated the effects of the frequency of visits on influencing practices and performances. For example, pregnant women screening for HIV increased significantly from the second visit in Nigeria [16], and the consistency in

pneumonia case management improved from 38 to 78 percent between the first to fourth supportive supervision in Ethiopia [7]. For malaria case management, the adjusted regression analysis showed that clinical performance against the checklist improved by an estimated six percentage point by the third visit [19]. Progress on most care steps for malaria case management were observed by having only one visit. However, palpable changes were observed when the supervision was structured in such a way that a second visit within 3 to 4 months was followed by a third at 12 months [19].

Moreover, facility characteristics such as; availability of electricity and water and distance from the woreda's capital city, and the organization of the supervision framework are also very important factors. For example, for projects establishing a supervision mechanism, the distance between the main station and the number of woredas assigned to each supervisor should be considered thoughtfully. The success of supervisions also depends on the quality of time spent between the supervisor and supervisees. Similarly, supervision that included supportive elements i.e. feedback and discussion of problems, was associated with quality to a fractionally greater degree than other supervision as shown in antenatal care through an increase of 0.10 standard deviation in quality score and in sick child care through an increase of 0.12 standard deviation in seven countries in sub-Saharan region [1].

Reading through the results from the study, it is good to note some of the limitations. The supervisions were made by project staff who may influence the observations. In addition, the analysis was only able to control the effect of background information which are available with the authors. A lack of baseline information about the study facilities and drop out of some of the facilities in due course of the program are the additional limitations of this study.

## Conclusions

Supervision is contributing to improvements made in the service delivery management at the health center level. Following a robust model for developing a supervision checklist and regular evaluation helps program managers, policy makers and other stakeholder take appropriate action promptly. As the influence of the supervisor may be limited to specific components, introducing a multidisciplinary team as well as engagement at various levels may facilitate quicker changes in the system as well as enhance supervision skills at the system level. Supervisory visits need to be structured to influence service delivery at primary health care as five visits each separated by 6 to 9 months to bring about significant service delivery improvements at the health center level. In addition, after the fourth visit, any checklist-based supervision needs to be transitioned to issue specific supportive supervision nested in overall quality improvement system.

## Supporting information

**S1 File.**
(DTA)

**S1 Data.**
(PDF)

## Acknowledgments

The authors thank Kristin Eifler and Heran Demissie for continued guidance and English language edit, respectively.

## Author Contributions

**Conceptualization:** Binyam Fekadu Desta.

**Data curation:** Ismael Ali Beshir, Habtamu Zerihun Demeke.

**Formal analysis:** Ismael Ali Beshir.

**Methodology:** Binyam Fekadu Desta, Ismael Ali Beshir, Bekele Belayhun Tefera, Mesele Damte Argaw.

**Project administration:** Binyam Fekadu Desta, Mengistu Asnake Kibret.

**Validation:** Mesele Damte Argaw, Mengistu Asnake Kibret.

**Writing – original draft:** Binyam Fekadu Desta.

**Writing – review & editing:** Ismael Ali Beshir, Bekele Belayhun Tefera, Mesele Damte Argaw, Habtamu Zerihun Demeke, Mengistu Asnake Kibret.

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
