## [Decision Letter · Decision Letter 0]

28 Apr 2020

PONE-D-20-05163

Does frequency of supportive supervisory visits influence health service delivery? – Dose and Response Study

PLOS ONE

Dear Mr Desta,

Thank you for submitting your manuscript to PLOS ONE. After careful consideration, we feel that it has merit but does not fully meet PLOS ONE’s publication criteria as it currently stands. Therefore, we invite you to submit a revised version of the manuscript that addresses the points raised during the review process.

Great job on a paper well done! The reviewers viewed it favorably, but have minor comments that will make for a clearer read. I hope you would strongly consider revising the submission based on their comments and resubmit for strong consideration for publication.

We would appreciate receiving your revised manuscript by Jun 12 2020 11:59PM. To enhance the reproducibility of your results, we recommend that if applicable you deposit your laboratory protocols in protocols.io, where a protocol can be assigned its own identifier (DOI) such that it can be cited independently in the future. For instructions see: http://journals.plos.org/plosone/s/submission-guidelines#loc-laboratory-protocols

We look forward to receiving your revised manuscript.

Kind regards,

Zhi Ven Fong, M.D., M.P.H.

Academic Editor

PLOS ONE

Journal Requirements:

2. Please refrain from stating p values as 0.00, either report the exact value or employ the format p<0.001.

3.  Please include additional information regarding the survey used in the study and ensure that you have provided sufficient details that others could replicate the analyses. For instance, if you developed a survey or check-list as part of this study and it is not under a copyright more restrictive than CC-BY, please include a copy, in both the original language and English, as Supporting Information.

5. Thank you for stating the following in the Competing Interests/Financial Disclosure* (delete as necessary) section:

"The funder had no role in study design, data collection and analysis, decision to publish, or preparation of the manuscript."

We note that one or more of the authors are employed by a commercial company: "USAID Transform: Primary Health Care project, JSI Training & Research Institute, Inc.,"

Reviewers' comments:

Reviewer's Responses to Questions

**Comments to the Author**

1. Is the manuscript technically sound, and do the data support the conclusions?

Reviewer #1: Yes

Reviewer #2: Yes

2. Has the statistical analysis been performed appropriately and rigorously? 

Reviewer #1: I Don't Know

Reviewer #2: Yes

3. Have the authors made all data underlying the findings in their manuscript fully available?

Reviewer #1: Yes

Reviewer #2: Yes

4. Is the manuscript presented in an intelligible fashion and written in standard English?

Reviewer #1: Yes

Reviewer #2: Yes

5. Review Comments to the Author

Reviewer #1: This is a well written manuscript evaluating the effects of a varying number and the ideal interval of described supervisory health center visits on the five service delivery components of the PHCPI framework, collected as an aggregate score. The authors use both descriptive analyses and a multilevel linear mixed model to build their analysis. The authors chose appropriate stratification variables in their analysis and establish a good description of their thought process throughout the paper.

A few comments:

1. Very minor grammatical errors including oxford comma placement throughout the manuscript.

2. In the Data Source section, it would be useful to understand definition of what a health center is. Are these akin to primary care doctors offices? This description will help readers understand the true setting of where the patients are receiving care.

3. Figure 1: in the service delivery section of the figure, only 4 components are listed where the paper refers to 5 components. It appears the community engagement component is not included in the figure.

4. How was it determined which facility received additional visits in the dataset?

5. Table 4, the upper CI appears to have been accidentally replaced by the p-values.

6. T value for 6-9 month visit in results section (page 16, line 2) seems to be incorrect as lower confidence interval is reported in the text, as opposed to the actual t-value.

7. Why are the beta values for the estimates of fixed effects reported in table 6 negative? It would be beneficial to readers to include a quick sentence describing how to interpret these beta values.

8. How was it determined that the visit between 6-9 months was the ideal interval?

I will recommend the paper for publication following the addressing of the above comments. I look forward to review this paper again and good work.

Reviewer #2: Thank you for the opportunity to review this interesting article (PONE-D-20-05163) by Desta and colleagues entitled, “Does frequency of supportive supervisory visits influence health service delivery? – Dose and Response Study.” This study aimed to determine the association of number and frequency of supervised healthcare visits and improvements in healthcare. To address this aim, the authors employed a retrospective cohort study design analyzing the outcome (Service-Delivery, PHCPI framework) over 3,080 supervised visits in a 2.5 year study period. Using a linear mixed model, the results demonstrated an improvement in service delivery over time until the 5th to 6th supervised visit. The authors therefore conclude that 5 supervised visits are associated with improved service delivery.

Specific Comments:

Abstract

1. Please further define the outcome variable (Service-Delivery) or PHCPI framework; this will assist the reader in interpreting the results.

2. In the statement in the results where the time interval of 6-9 months “showed more significant contribution”; please include the results of the statistical test that support this finding

Introduction

1. Comprehensive and nice introduction to the formal definition of “supervision,” and also what is currently unknown in the literature (association of supervision on improving outcomes). The authors could expand here, specifically regarding what outcomes they are referring to (e.g., process measures [number of visits, compliance metrics, etc.] or patient outcomes, or both).

2. Did the authors have an a priori hypothesis on the dose-response association (number of supervised visits and service delivery quality) or frequency of visits?

Methods

1. Study setting: please clarify if 25,000 people/health center is the overall estimated volume or estimated annual volume. This will help the reader appreciate the context of your findings in terms of number/frequency of visits.

2. Intervention: recommend the authors include a representative supportive supervision checklist as an appendix. The 30 questions can be included.

3. Any sense of the quality of the supervision – who are the supervisors? What is their interaction during the visit? Is there potential for bias/Hawthorne effect?

4. What types of visits and health centers were included/excluded, if any?

5. Study design and instruments: please expand upon why the Service-Delivery component of the PHCPI framework was selected as the outcome of the study. The authors may consider referencing the introduction’s definition of quality care (accessibility, abilities/motivation of a workforce) and further expanding on the PHCPI conceptual frameworks definition of Service-Delivery. The way this composite measure is calculated is appropriately described in the “Outcome variables” section of the Methods, however further detail on the underlying construct in the PHCPI framework would enhance the reader’s understanding.

6. The authors appropriate describe the selected statistical/quantitative methods based on the structure and missingness of the data.

Results

1. Facility characteristics: did any of the included facilities have prior supervised visits before the onset of this study? Or, do the number of study visits (ranging from 1-7 based on the results) refer to the number of visits during the study period of July 2017-December 2019?

2. Is there any baseline performance data available for the facilities, either directly measured Service-Delivery or a surrogate?

3. Are there any characteristics of the facilities that had a greater number of visits that differ from those that only had one visit? Were certain facilities scheduled to receive multiple visits? Does the observed improvement in service-delivery reflect bias, in that those facilities who had a greater number of visits were already high-performers? In reading Table 2, as an example, it appears that there were increasing proportions of facilities that had “Access to Roads” and “Has at least one ambulance;” is this because infrastructure improvements occurred at these facilities in the study time period, or because those facilities that did not meet these criteria had dropped out over time?

Discussion

1. The discussion is well-written and appropriately contextualizes the results from this analysis with existing literature.

2. The limitations section should be expanded to address the heterogeneity in the number of visits at each facility/attrition of certain facilities and the unknown baseline assessments of service delivery at the included facilities.

6. PLOS authors have the option to publish the peer review history of their article (what does this mean?). If published, this will include your full peer review and any attached files.

Reviewer #1: No

Reviewer #2: No

---

## [Author Response · Author response to Decision Letter 0]

15 May 2020

PONE-D-20-05163

Does frequency of supportive supervisory visits influence health service delivery? – Dose and Response Study

PLOS ONE

Journal Requirements:

Response: 

All the guidance in the above sources were reviewed and necessary correction in the document has been made. 

2. Please refrain from stating p values as 0.00, either report the exact value or employ the format p<0.001.

Response: 

Corrected as suggested. 

3. Please include additional information regarding the survey used in the study and ensure that you have provided sufficient details that others could replicate the analyses. For instance, if you developed a survey or check-list as part of this study and it is not under a copyright more restrictive than CC-BY, please include a copy, in both the original language and English, as Supporting Information.

Response: 

The extracted tool from the online form is uploaded as a supplemental file. 

Response:

Agreed to upload the minimally anonymized data as a supplemental file. 

Responses: 

The minimally anonymized data is uploaded as a supplemental file and the cover letter has been updated to reflect this. 

5. Thank you for stating the following in the Competing Interests/Financial Disclosure* (delete as necessary) section:

Response

As suggested the following statement is inserted. 

“The funder provided support in the form of salaries for authors [insert relevant initials], This does not alter our adherence to PLOS ONE policies on sharing data and materials. The funder did not have any additional role in the study design, data collection and analysis, decision to publish, or preparation of the manuscript. The specific roles of these authors are articulated in the ‘author contributions’ section.”

Response: 

Corrected as suggested. 

Response: 

Corrected as suggested. 

Reviewer’s Comments to the Author

Reviewer #1: This is a well written manuscript evaluating the effects of a varying number and the ideal interval of described supervisory health center visits on the five service delivery components of the PHCPI framework, collected as an aggregate score. The authors use both descriptive analyses and a multilevel linear mixed model to build their analysis. The authors chose appropriate stratification variables in their analysis and establish a good description of their thought process throughout the paper.

A few comments:

1. Very minor grammatical errors including oxford comma placement throughout the manuscript.

Response: 

Corrected as suggested. 

2. In the Data Source section, it would be useful to understand definition of what a health center is. Are these akin to primary care doctors’ offices? This description will help readers understand the true setting of where the patients are receiving care.

Responses:

A description of the health center was already included in the study setting section of the manuscript. Additional clarification has been included in the study setting section. 

3. Figure 1: in the service delivery section of the figure, only 4 components are listed where the paper refers to 5 components. It appears the community engagement component is not included in the figure.

Responses: 

The figure with five components is included in the revised document. Necessary changes in the descriptions have also been made. 

4. How was it determined which facility received additional visits in the dataset?

Responses: 

Each district and facility in the DHIS2 system has a unique identifier. The number of times those identifiers pop up in the system reflects frequency of visit. Based on this information, the data set includes order of visits/frequency of visits. 

5. Table 4, the upper CI appears to have been accidentally replaced by the p-values.

Responses:

The typing error is corrected as suggested. 

6. T value for 6-9 month visit in results section (page 16, line 2) seems to be incorrect as lower confidence interval is reported in the text, as opposed to the actual t-value.

Responses: 

The typing error is corrected as suggested

7. Why are the beta values for the estimates of fixed effects reported in table 6 negative? It would be beneficial to readers to include a quick sentence describing how to interpret these beta values.

Responses: 

Explanation of the beta effect is included in the result section. The last visit is used as the reference point to demonstrate higher performances. Any comparison against the previous visits leads to negative coefficients. 

8. How was it determined that the visit between 6-9 months was the ideal interval?

Responses: 

The statistical analysis result showed any visit which happens before nine months may lead to improvements. However, the changes observed in the 6-9 month group is higher than the earlier visits. 

Reviewer #2: Thank you for the opportunity to review this interesting article (PONE-D-20-05163) by Desta and colleagues entitled, “Does frequency of supportive supervisory visits influence health service delivery? – Dose and Response Study.” This study aimed to determine the association of number and frequency of supervised healthcare visits and improvements in healthcare. To address this aim, the authors employed a retrospective cohort study design analyzing the outcome (Service-Delivery, PHCPI framework) over 3,080 supervised visits in a 2.5 year study period. Using a linear mixed model, the results demonstrated an improvement in service delivery over time until the 5th to 6th supervised visit. The authors therefore conclude that 5 supervised visits are associated with improved service delivery.

Specific Comments:

Abstract

1. Please further define the outcome variable (Service-Delivery) or PHCPI framework; this will assist the reader in interpreting the results.

Responses:

A sentence to address the feedback is included in the abstract section. 

2. In the statement in the results where the time interval of 6-9 months “showed more significant contribution”; please include the results of the statistical test that support this finding

Responses: 

A sentence to address the feedback is included in the abstract section. 

Introduction

1. Comprehensive and nice introduction to the formal definition of “supervision,” and also what is currently unknown in the literature (association of supervision on improving outcomes). The authors could expand here, specifically regarding what outcomes they are referring to (e.g., process measures [number of visits, compliance metrics, etc.] or patient outcomes, or both).

Responses: 

Expanded as per the suggestions. 

2. Did the authors have an a priori hypothesis on the dose-response association (number of supervised visits and service delivery quality) or frequency of visits?

Responses: 

As we didn’t have such hypothesis, the study was aims to determine the number of visits that influence service delivery for future programming and fills the knowledge gap for the optimal number of visits that bring about changes. 

Methods

1. Study setting: please clarify if 25,000 people/health center is the overall estimated volume or estimated annual volume. This will help the reader appreciate the context of your findings in terms of number/frequency of visits.

Responses: 

The number only signifies the potential health service beneficiaries. Some of them may visit more than once. 

2. Intervention: recommend the authors include a representative supportive supervision checklist as an appendix. The 30 questions can be included.

Responses: 

We have included the checklist as a supplemental file.

3. Any sense of the quality of the supervision – who are the supervisors? What is their interaction during the visit? Is there potential for bias/Hawthorne effect?

Responses: 

Descriptions have been included in the document.

4. What types of visits and health centers were included/excluded, if any?

Responses: 

All health centers under a project support area were included in the study. 

5. Study design and instruments: please expand upon why the Service-Delivery component of the PHCPI framework was selected as the outcome of the study. The authors may consider referencing the introduction’s definition of quality care (accessibility, abilities/motivation of a workforce) and further expanding on the PHCPI conceptual frameworks definition of Service-Delivery. The way this composite measure is calculated is appropriately described in the “Outcome variables” section of the Methods, however further detail on the underlying construct in the PHCPI framework would enhance the reader’s understanding.

Responses: 

Corrected as suggested. 

Results

1. Facility characteristics: did any of the included facilities have prior supervised visits before the onset of this study? Or, do the number of study visits (ranging from 1-7 based on the results) refer to the number of visits during the study period of July 2017-December 2019?

Responses: 

The Authors do not have any information about a prior visits before the project period. 

2. Is there any baseline performance data available for the facilities, either directly measured Service-Delivery or a surrogate?

Responses: 

No baseline was taken which has been included in the ‘limitation’ section. 

3. Are there any characteristics of the facilities that had a greater number of visits that differ from those that only had one visit? Were certain facilities scheduled to receive multiple visits? Does the observed improvement in service-delivery reflect bias, in that those facilities who had a greater number of visits were already high-performers? In reading Table 2, as an example, it appears that there were increasing proportions of facilities that had “Access to Roads” and “Has at least one ambulance;” is this because infrastructure improvements occurred at these facilities in the study time period, or because those facilities that did not meet these criteria had dropped out over time?

Responses: 

We conducted a test, (multinomial logistics regression) to check for the influence of access to roads on number of supervision visits and found no relationship between them. We do not anticipate significant change in accessibility to roads within this period. However, the authors believe that other factors such as number of ambulances may change over the course of supervision support. 

Discussion

1. The limitations section should be expanded to address the heterogeneity in the number of visits at each facility/attrition of certain facilities and the unknown baseline assessments of service delivery at the included facilities.

Responses: 

As suggested not knowing the baseline information and attrition of facilities are included as limitations. 

---

## [Decision Letter · Decision Letter 1]

3 Jun 2020

Does frequency of supportive supervisory visits influence health service delivery? – Dose and Response Study

PONE-D-20-05163R1

Dear Dr. Desta,

We’re pleased to inform you that your manuscript has been judged scientifically suitable for publication and will be formally accepted for publication once it meets all outstanding technical requirements.

Kind regards,

Zhi Ven Fong, M.D., M.P.H.

Academic Editor

PLOS ONE

Additional Editor Comments (optional):

Reviewers' comments:

Reviewer's Responses to Questions

**Comments to the Author**

1. If the authors have adequately addressed your comments raised in a previous round of review and you feel that this manuscript is now acceptable for publication, you may indicate that here to bypass the “Comments to the Author” section, enter your conflict of interest statement in the “Confidential to Editor” section, and submit your "Accept" recommendation.

Reviewer #1: All comments have been addressed

Reviewer #2: (No Response)

2. Is the manuscript technically sound, and do the data support the conclusions?

Reviewer #1: Yes

Reviewer #2: Yes

3. Has the statistical analysis been performed appropriately and rigorously? 

Reviewer #1: Yes

Reviewer #2: Yes

4. Have the authors made all data underlying the findings in their manuscript fully available?

Reviewer #1: Yes

Reviewer #2: Yes

5. Is the manuscript presented in an intelligible fashion and written in standard English?

Reviewer #1: Yes

Reviewer #2: Yes

6. Review Comments to the Author

Reviewer #1: You have addressed all of the comments requested. I am happy to support this manuscript for publication.

Reviewer #2: Thank you for the revisions. There is one area of the paper where interpretation of data remains ambiguous. This concerns attrition of facilities. Specifically, the second paragraph of the results section report that 100% of facilities had one visit, 29% had two visits, 14% had three visits, 5% had four visits, and 3 had five+ visits. The authors' key interpretation of the results from the rest of the analysis is that there is a dose-response association between number of supervised visits and Service-Delivery indicator. However, the reader is forced to wonder if the facilities that had a greater number of visits were high performers to begin with. There are two ways to address this for the reader. The first option is to produce another table like the current Table 1, which provides characteristics of the facilities that had one, two, three, four, five+ visits. This will help the reader determine if there are meaningful differences between these facilities. The second way is to perform a sensitivity analysis to determine if the Service-Delivery outcome changes with increased number of visits only among those groups that had multiple visits. Without this, it is difficult to support the conclusion of a true dose-response relationship, and the authors therefore would need to temper the stated conclusions and expand upon this limitation. Otherwise, thank you for addressing all the other revisions.

7. PLOS authors have the option to publish the peer review history of their article (what does this mean?). If published, this will include your full peer review and any attached files.

Reviewer #1: No

Reviewer #2: No

---

## [Editor Report · Acceptance letter]

4 Jun 2020

PONE-D-20-05163R1 

Does frequency of supportive supervisory visits influence health service delivery? – Dose and Response Study 

Dear Dr. Desta:

I'm pleased to inform you that your manuscript has been deemed suitable for publication in PLOS ONE. Congratulations! Your manuscript is now with our production department. 

Kind regards, 

on behalf of

Dr. Zhi Ven Fong 

Academic Editor

PLOS ONE